# Follow the Path: Hierarchy-Aware Extreme Multi-Label Completion for Semantic Text Tagging

## ABSTRACT

Extreme Multi Label (XML) problems, and in particular XML completion – the task of prediction the missing labels of an entity – have attracted significant attention in the past few years. Most XML completion problems can organically leverage a label hierarchy, which can be represented as a tree that encodes the relations between the different labels. In this paper, we propose a new algorithm, HECTOR – Hierarchical Extreme Completion for Text based on transfORmer, to solve XML Completion problems more effectively. HECTOR operates by directly predicting *paths in this tree*, instead of simple labels, thus taking advantage of information encoded in the hierarchy. Due to the sequential aspect of these paths, HECTOR can leverage the effectiveness and performance of the Transformer architecture to outperform state-of-the-art of XML completion methods. Extensive evaluations on three real-world datasets demonstrate the effectiveness of our approach for XML completion. We compare HECTOR with several state-of-the-art XML completion methods for various completion problems, and in particular for label refinement, i.e., the scenario where only the coarse labels (i.e. the first few top levels in a taxonomy) are observed. Empirical results on three different datasets show that our method significantly outperforms the state of the art, with HECTOR frequently outperforming previous techniques by more than 10% according to multiple metrics.

## CCS CONCEPTS

• **Computing methodologies** → **Supervised learning by classification**.

## KEYWORDS

Semantic Tagging, Taxonomy, Extreme Multi-Label Classification, Label Completion, Transformers

**ACM Reference Format:**
Anonymous Author(s). 2024. Follow the Path: Hierarchy-Aware Extreme Multi-Label Completion for Semantic Text Tagging . In *TheWebConf2024: The ACM Web Conference 2024, May 13–17, 2024, Singapore*. ACM, New York, NY, USA, 13 pages. https://doi.org/10.1145/1122445.1122456

## 1 INTRODUCTION

As the number of textual documents has grown exponentially over the past decades, [7], Multi-Label text Classification (MLC), which is the task of assigning the most relevant subset of labels to documents, has received significant attention [17, 34, 37]. Indeed, MLC is able to represent the semantic contents of a document using a series of key concepts (also known as *semantic tags*), which in turn eases the organization of information and helps users navigate large text collections. MLC is for instance key for online scientific literature: as the number of scientific papers getting published online is rapidly increasing, semantic tagging becomes crucial to support the discovery of new scientific results as well as exploratory efforts within and across fields of interest [31].

The number of potential labels has also increased dramatically – collections of thousands to tens of thousands of labels are now routinely used to tag documents – and new dedicated methods called Extreme Multi-Label text Classification (XMLC) [34] have been developed. Indeed, XMLC poses additional computational challenges due to the large number of labels and the uneven distribution of their occurrences, typically leading to a long tail of rare labels.

XMLC problems can typically leverage a label hierarchy, since managing large collections of labels is in itself a challenge in real world settings. For scientific literature, there exist for instance many well-designed hierarchical ontologies and taxonomies of concepts, which can be used as hierarchy of labels for scientific text tagging [37]. Today, some of the most popular ontologies in that context include the ACM CCS[1], a poly-hierarchical ontology containing concepts related to Computer Science, the MeSH thesaurus[2], which was developed to index and search biomedical and health related data, and the Microsoft Academic Graph (MAG) [31], which provides a taxonomy of concepts from different domains. These hierarchies impose natural partial orders on labels, from more general to more specific, and can provide valuable information for XMLC tasks. Following this observation, several recent studies have proposed approaches to embed this meta-information into the XMLC problem. Notably, [5] proposed to learn an embedding of the label space by first performing a clustering of labels using their short descriptions, and thus reducing the complexity of the output space, while [37], leveraged metadata by modifying the loss function to force proximity in the joint embedding space.

An interesting sub-problem of XMLC that is the focus of this work is Extreme Multi-Label Completion (XMLCo), where each document instance is already tagged with a *partial* set of labels that the model has to complete, by leveraging both the document content as well as existing labels. The problem of incomplete labels is frequently encountered in many application domains due to multiple compounding factors, including the subjectivity of human annotators, time-dependent data, the addition of new sub-concepts as leaves to a taxonomy, time constraints, or privacy concerns. Label completion plays a crucial role in enhancing the completeness and accuracy of datasets [25]. This sub-problem is particularly relevant

[1]https://dl.acm.org/ccs
[2]https://www.nlm.nih.gov/mesh/meshhome.html

in the context of the taxonomies, as they frequently abide by the hierarchy constraint [26], i.e. taxonomies that can be represented as trees. Indeed, it has been observed that in this case, data instances are equipped with general, high level labels, while more specific labels are more often missing [25]. We refer to the task of adding more specific label to a data instance as Label Refinement.

In the present work, we introduce a new Transformer-based encoder-decoder model for XMLCo, named HECTOR[3] (Hierarchical Extreme Completion for Text based on transfORmer), which directly takes advantage of the hierarchical structures of the label space to better predict missing labels and solve Label Refinement. Transformers [30] have demonstrated state-of-the-art results on many NLP-related tasks, such as document summarization, text generation, or named entity recognition [29], and in particular have been successfully applied to XMLC [5, 13]. However, to the best of our knowledge, previous applications focused on the encoder part of the original Transformer architecture [5], or the label were predicted as an unstructured set (see [13] and reference therein), as labels do not intrinsically possess a sequence structure. Conversely, our technique HECTOR fully leverages the sequence-to-sequence (Seq2Seq) nature of transformers, by predicting *paths* in the hierarchy of labels. This approach has two significant advantages:

(1) HECTOR benefits from the performance of transformer on Seq2Seq tasks, which have been proven to be very effective for MLC tasks (Raffel et al., 2020; Chung et al., 2022).

(2) HECTOR organically leverages all the meta-information contained in the hierarchical tree organizing the labels, without needing to learn or approximated it through pre-training or regularization.

We evaluate the effectiveness of our approach through a wide range of experiments of label completion, with particular focus on Label Refinement – the case of label completion where *general* labels are provided, i.e. labels representing broader categories or higher-level concepts in the hierarchy. Our evaluation results highlight the advantage of HECTOR over existing methods, and show that it significantly outperforms other methods for label refinement on a wide range of metrics and on three datasets, with HECTOR frequently outperforming previous techniques by more than 10% according to multiple metrics.

The rest of the paper is organized as follows. In Section 2, we provide some background information and review related works on XMLCo. We present our approach and HECTOR's architecture in Section 3. Section 4 introduces the baselines and presents our experimental results.

## 2 BACKGROUND AND RELATED WORK

**Extreme Multi-Label Classification** Traditional MLC approaches can be divided into three groups: one-vs-all, embedding-based and tree-based methods [16]. *One-vs-all* methods independently train a binary classifier for each label. In extreme settings with thousands of labels, this approach can be prohibitively expensive. To reduce training complexity and model size, different techniques were proposed, among them margin-maximizing loss with $l_1$ penalty [33], parameter thresholding [1], label filtering [19], learned label trees [14, 22] and negative sampling [11]. *Tree-based* methods recursively

partition the instance set or the label set and at each non-leaf node train a classifier focusing on a small subset of the original large-scale problem [12, 21, 23]. *Embedding* methods aim at learning the latent low dimensional vector space of the labels, and perform classification by finding the nearest label neighbors for each test instance [2, 10, 28]. Closer to the present work, there have been a growing number of works demonstrating the efficiency of *deep learning* for the XMLC task in the last few years. XML-CNN [15] is one of the pioneers in this area, proposing to apply a convolutional neural network (CNN) to learn the text representation. More recently, [34] introduced AttentionXML, which leveraged a multi-label attention mechanism and shallow probabilistic label trees (PLT). X-Transformer [5] was the first attempt to fine-tune deep Transformer models to the XMLC task, and was then further improved by [36] through the use of recursive fine-tuning. More recently, [13] analyzed different types of Transformer-derived architecture for the XMLC task, and show that model using a seq2seq approach tend to perform better – a prime motivation behind HECTOR. Furthermore, compared to these methods, HECTOR is able to efficiently leverage the hierarchical taxonomy of the labels.

**Hierarchical Multi-Label Classification** Hierarchical classifiers have long been used in MLC [3], and recent works have proposed strategies to enhance XMLC methods using the structure of labels. [8] proposed to incorporate the tree of labels directly into the architecture of the neural network, while Gargiulo et al. [9] proposed a convolutional neural network to address this task. More recent work has combined ideas from Hierarchical Multi-Label and transformers. MATCH [37] used hierarchical relations among labels for regularization, enforcing each label to be similar to its parents, while Caled et al. [3] introduced a recurrent neural network with a hierarchical output layer, where each deeper level gets predictions from the previous levels as an additional input. However, to the best of our knowledge, HECTOR is the first to predict a path directly following the hierarchy of labels, thus combining the Seq2Seq strengths of transformers with Hierarchical Multi-Label strategies.

**Label Completion** Many label completion techniques rely on matrix completion, where the correlation between labels occurrences is used to predict missing labels. For instance, [6] proposed an approach that uses both local and Global attention to improve Matrix completion. However these methods generally do not scale to XMLCo problems, due to the size of the dataset and the number of labels. More recently, Romero et al. [25] used a hierarchical approach to complete the annotation of genes with biological functions. They first train a global classifier which predicts probabilities of each label independently, and then aggregate these probabilities along the path in a hierarchical label tree to compute final probabilities for leaf labels. Compared to these methods, HECTOR uses the performance of transformers and directly embeds the label tree by predicting paths on this tree.

## 3 METHOD

### 3.1 Intuition

We begin by introducing the intuition behind HECTOR, and the use of transformers for XMLCo problems.

**First**, positive labels assigned to a document are usually represented by specific tokens in an input document. Figure 1 illustrates

---
[3]We will release the code of the model upon acceptance

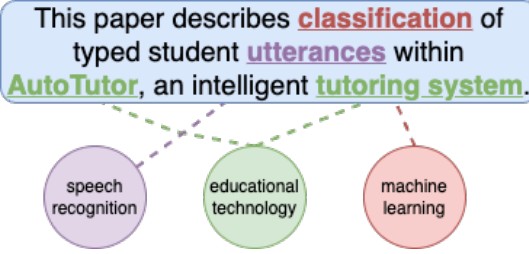

**Figure 1: Token-label interaction: highlighted text tokens correspond to different labels (grouped by color)**

the idea. Transformers, through their cross-attention mechanism, are able to take into account fine-grained dependencies between tokens, and focusing on the most relevant parts of the input sequence with respect to each label. The advantages of this approach has been demonstrated by previous work, that used attention mechanisms and transformers to achieve state-of-the-art performance on XMLCo, such as AttentionXML [34] and XR-Transformer [36].

**Second**, the available label hierarchy contains valuable information that can be used for XMLCo. For example, the presence of label *SQL* can be a strong indicator of relevance for label *RDBMS* and vice versa. This idea is at the heart of many successful label completion approaches such as [25]. A notable previous approach to label correlation modeling was proposed in [24], where authors constructed a chain of binary classifiers (one for each label) and where the output of each following classifier was conditioned on outputs of all previous classifiers. Interestingly, this multi-label classification approach with a chain of classifiers is similar to the decoding process in a Seq2Seq model, where an output sequence is generated one token at a time, with each subsequent token being conditioned on the previously generated tokens.

HECTOR combines these two ideas, by using a novel paradigm for multi-label completion: instead of predicting *individual* labels, it predicts *paths* in a label tree.

## 3.2 Path Prediction

In the rest of this paper, we assume that labels are organized hierarchically, e.g., in a taxonomy. In particular, we assume that the taxonomy abides by the hierarchy constraint [25], and therefore can be represented as a tree. Hereinafter we will use the terms *taxonomy* and *label tree*[4] interchangeably to refer to the hierarchical label structure. Using the taxonomy, we model a set of labels assigned to a document as a set of paths in a label tree, as shown in Figure 2. As opposed to a set of labels, each path *does* naturally yield a sequence structure, and thus can be used in Seq2Seq models.

*Path Completion.* It is important to note that while many XMLC datasets may abide by the hierarchy constraint [25], the set of labels assigned to each document may be incomplete, e.g. they do not constitute complete paths in the tree. This is due to the fact that labels are sometimes assigned inconsistently: for instance, in some cases only leaves are included, whereas in other cases top-level labels and some leaves are included, but not all labels in the

---

[4]A taxonomy can also have a graph structure, but within the present research we focus on trees and leave more complex data structures for future work.

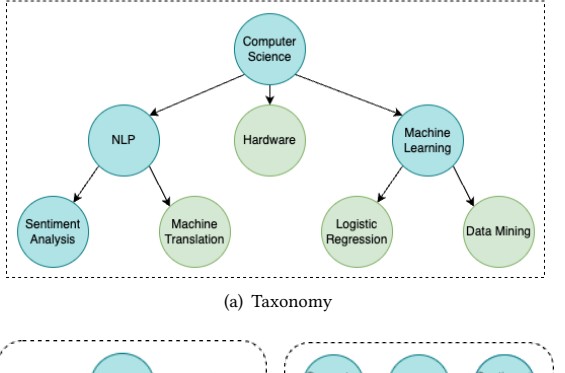

(a) Taxonomy

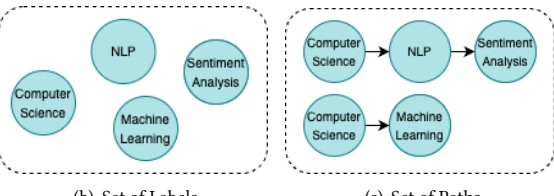

(b) Set of Labels          (c) Set of Paths

**Figure 2: Converting a set of labels into a set of paths leveraging a label hierarchy**

middle of the paths. We thus complete the label sets for each data point by adding all the missing ancestors to each label in order to obtain coherent paths, similarly to Hierarchical Label Set Expansion proposed by [9]. Formally, we proceed as follows:

- For each label $l_j$ from the original label set $\mathcal{L}$, we build a path $p_j$ from the root of the label tree to $l_j$.
- We update $\mathcal{L}$ with labels $l_j^k \in p_j$, if $l_j^k \notin \mathcal{L}$.

Throughout this paper, we operate on datasets modified as described above, i.e., with positive label sets extended to contain full paths in a label tree. For example, using a toy taxonomy from Figure 2(a), if the original label set consists of labels $\mathcal{L} =\{$ NLP, Logistic Regression$\}$, its completed version will be $\mathcal{L}' =\{$NLP, Logistic Regression, Machine Learning$\}$[5].

To summarize, we reformulate a multi-label completion task as a path decoding task, which can be outlined as follows:

(1) **Preprocessing**: complete and regroup positive labels assigned to each document to form a set of paths in the tree.
(2) **Training**: train a Seq2Seq model, where a document is an input sequence, and a path is a target sequence.
(3) **Inference**: given an input document and an incomplete set of labels, decode path(s) in the label tree. they are then merged and sorted by label scores to generate a final ranking of labels, which is then used for prediction.

One additional advantage of our approach is that labels in a path are decoded sequentially, from the most general concepts (first level of the taxonomy) to more specific concepts. We argue that this approach is particularly well suited for label completion, as illustrated by our experiment results (see Section 4).

---

[5]We do not add the root label to the label set since it is trivial to predict.

## 3.3 Model Architecture

In this subsection, we introduce HECTOR – a **H**ierarchical **E**xtreme **C**ompletion for **T**ext Based on Transf**OR**mer. HECTOR's architecture is based on Transformers [30] – the last generation Seq2Seq model, which proved extremely efficient on several NLP tasks. Similar to previous Seq2Seq models, Transformers utilize an encoder-decoder architecture, but they dispenses with recurrence and convolutions relying entirely on the attention mechanism to compute representations of their input and output. More specifically, Transformers feature the following types of attention:

- Encoder self-attention: enables the encoder to consider the context of each word based on the entire *input* sequence.
- Decoder self-attention: allows the decoder to consider the influence of previously generated tokens on the current token generation step.
- Encoder-decoder cross-attention: allows the decoder to focus on relevant parts of the encoder's output during the generation process.

In the context of our task, these three types of attention perform the following functions: the encoder self-attention learns contextualized embeddings of tokens in the input document; the encoder-decoder cross-attention captures fine-grained dependencies between input tokens and output labels; the decoder self-attention considers previously predicted labels to generate a coherent path in the label tree. HECTOR's architecture is outlined in Figure 3. In the following, we introduce the main components of Transformers as well as the key changes we made to the original architecture to adapt it to our setting.

**Encoder.** The encoder in the Transformer model extracts features from the input sequence, enabling the model to capture the relationships between the input tokens and create rich representations for further processing by the decoder. The encoder is composed of a stack of $N = 6$ identical layers. Each layer consists of a multi-head self-attention mechanism and a fully connected feed-forward network with a residual connection. In HECTOR's encoder, we mostly follow the original Transformer architecture with some specific changes. We use pre-trained GloVe embeddings [20] as our initial word representation, hence both encoder input and output are 300-dimensional. For this reason, we also changed the number of attention heads from 8 to 12 (as a rule of thumb, model dimension should be dividable by the number of heads).

**Decoder.** The decoder in the Transformer model takes the encoded input and uses attention mechanisms to generate a coherent output sequence, capturing contextual relationships between the generated tokens. During training, the decoder takes the ground-truth output sequence in addition to the encoder output to learn dependencies between output tokens – this algorithm is referred to as *teacher forcing*. During inference, the decoder takes the encoder output and generates the output sequence from scratch, one token at a time. As the encoder, the decoder is composed of a stack of $N = 6$ identical layers, with an additional encoder-decoder cross-attention block at each layer. As opposed to traditional Seq2Seq tasks, where both input and output sequences consist of words, in our case the output is a sequence of labels. In natural language there are synonymous words that are semantically similar, therefore their embeddings can be very close to each other in the vector space. On the other

hand, in the label space all embeddings should be clearly separated, as we assume that there are no semantically similar labels. For better distinguishability, we increase the dimensionality of label embeddings from $d = 300$ to $d = 600$. Label embeddings are initialized randomly and learned during the training phase. Since in the Transfomer model the encoder output and decoder input should be of the same dimension, we add an additional fully connected layer between the encoder and the decoder, which performs dimensionality expansion. We refer to this component as the *adapter*. We empirically investigate the effect of the increased dimensionality of label embeddings in Section 4.3.

**Prediction Layer.** The decoder generates contextualized label representations, which are projected onto final $|V|$-dimensional vectors, where $|V|$ is the size of the label vocabulary. Each element of the resulting vector represents the probability of the corresponding label. The prediction layer consists of a fully-connected layer followed by a Softmax activation function.

**Loss Function.** Following the original Transformer architecture, we use the Kullback-Leibler divergence loss, which measures the dissimilarity between two probability distributions. During training, we use label smoothing of value $\epsilon_{ls} = 0.2$ [27]. Label smoothing is a regularization technique, which involves replacing the one-hot encoding of the target labels with a smoothed distribution. Instead of assigning a probability of 1 to the true label and 0 to all other labels, label smoothing assigns a confidence score to the true label and redistributes the smoothing mass among the other labels. In HECTOR, we introduce some prior knowledge about the label taxonomy into the loss function. Since we aim at decoding tree paths rather than unstructured sequences, we know in advance which labels can occur at each position. Thus, at the $i$-th position only labels from the $i$-th level (i.e., at depth $i$) of the taxonomy can appear. We leverage this knowledge by applying a mask onto the labels, such that the smoothing mass is redistributed on the corresponding level, setting the probabilities of all other labels to 0. We discuss the impact of this approach in Section 4.3.

**Training.** In multi-label problems, each document can have labels from different (sub)-domains, resulting in multiple paths in the label tree. During training, we randomly select one path per document as the ground-truth for each training epoch. The idea behind this approach is to introduce some variability during training and avoid overfitting to a specific output sequence – in line with the observations of [32]. By randomly selecting one of the possible output paths as the ground-truth during training, the model learns to generate all the possible output paths with equal probability.

## 3.4 Label Completion with HECTOR

At inference time, the prefix of known labels is provided to HECTOR. Then, we use beam search to generate multiple paths for each data point and to predict missing labels. As opposed to greedy search, where at each step a candidate with the highest probability is selected and passed to the next step, the beam search algorithm maintains a *set* of the most promising candidate sequences, known as the beam. More formally, the beam search algorithm proceeds as follows:

- The model generates a set of candidate labels for position $i$.

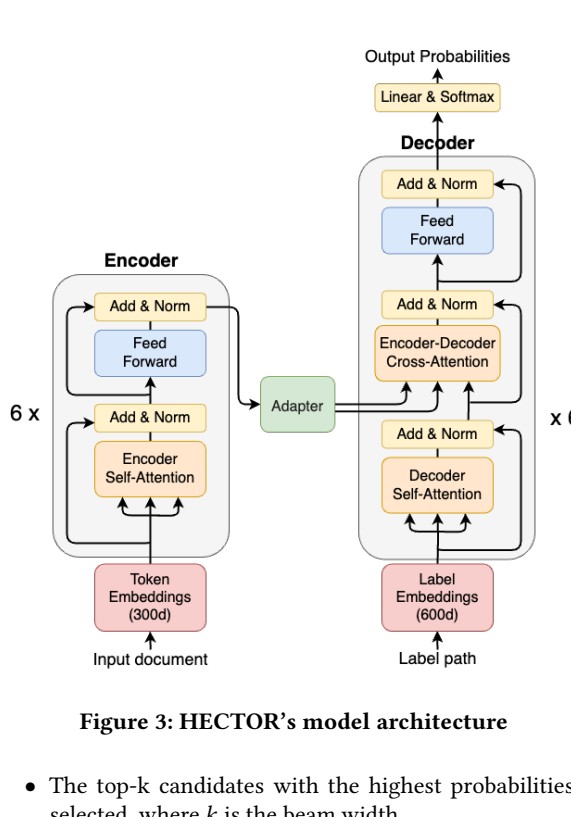

**Figure 3: HECTOR's model architecture**

- The top-k candidates with the highest probabilities are selected, where $k$ is the beam width.
- The selected candidates are appended to preceding partial sequences (predicted labels from positions 1 through $i - 1$) and joint probabilities of *extended* sequences are computed.
- The top-k extended sequences are passed to the next step for generating a set of candidate labels for position $i + 1$.

The beam search algorithm aims at maximizing probabilities of full sequences rather than individual elements of a sequence. Additionally, it allows decoding multiple sequences simultaneously, which is important in the context of our task since each document may have multiple relevant label sequences.

After performing beam search, we merge all decoded paths in a flat list and sort labels by their individual probabilities to produce the final ranking. Formally, we proceeded as follows. For a document $d$, let $\mathbf{P}(l_j | l_1, \ldots l_{j-1})$ denote the predicted probability of observing label $l_j$ given the path $l_1, \ldots l_j - 1$. We compute the path-independent marginal probability of the label $l_j$ as

$$\mathbf{P}(l_j) = \max_{\text{possible paths } l_1, \ldots, l_{j-1}} \left( \prod_{i=1}^{j} \mathbf{P}(l_i | l_1, \ldots l_{i-1}) \right)$$

In other words, we take the maximum probability of the label occurring across all possible paths in the taxonomy.

## 4 EXPERIMENTAL EVALUATION

In this section, we extensively evaluate HECTOR on label refinement tasks through multiple experiments. As introduced above, label refinement is an important task in practice (as new concepts are typically appended as leaves in the taxonomy) and a special case of label completion, where documents are labeled with general concepts (corresponding to the first level(s) of a label hierarchy),

**Table 1: Dataset statistics**

|  | $N_{train}$ | $N_{test}$ | $L$ | $\overline{L}$ | $\overline{P}$ | $\overline{W}$ | $H$ |
|---|---|---|---|---|---|---|---|
| **MAG-CS** | 89,920 | 54,008 | 2,641 | 4.4 | 2.9 | 87 | 6 |
| **PubMed** | 100,042 | 39,890 | 5,911 | 18.5 | 3.3 | 142 | 15 |
| **EURLex** | 45,000 | 6,000 | 4,492 | 10.4 | 4.9 | 288 | 7 |

$N_{train}$: #training instances, $N_{test}$: #test instances, $L$: #labels, $\overline{L}$: average #labels per instance, $\overline{P}$: average #paths per instance, $\overline{W}$: average #words per instance, $H$: height of the label tree.

and the algorithm is tasked to predict more specific (lower level) concepts. The exact nature of the task depends on the level $L$, from which we start the refinement process, i.e., we assume that labels from level 1 to $L-1$ are observed. Interestingly, since the taxonomies we study abide by the hierarchy constraint and are complete (see Section 3.2), all label completion tasks can be seen as label refinement, since predicting general labels given specific labels is trivial in this setting. Furthermore, the XML classification task can be seen as a specific case of label refinement with $L = 1$ (since the root of the tree is common to all data points, and therefore does not bring any information).

### 4.1 Experimental Setting

**Datasets.** We evaluate our method on three well-known and large-scale datasets: MAG-CS, PubMed and EURLex. We report important statistics from our datasets in Table 1, and Figure 4 summarizes label distribution per level in 3 datasets.

- **MAG-CS.** The Microsoft Academic Graph (MAG) Computer Science (CS) is a subset of the MAG dataset [31] focused on the computer science domain, containing papers published at 105 top CS conferences from 1990 to 2020, while the label tree contains relevant concepts descendants of the root-level "Computer Science". [37]
- **PubMed.** we use a subset of PubMed released by [37], which comprises papers published in 150 top journals in medicine from 2010 to 2020. Each PubMed paper is labeled with relevant concepts from the Medical Subject Headings (MeSH) hierarchically-organized thesaurus.
- **EURLex.** EURLex [18] is one of the most common XMLC benchmark datasets. It contains English EU legislative documents from the EUR-LEX portal[6], tagged with concepts (labels) from the European Vocabulary (EuroVoc)[7]. We use the latest version of EURLex released by [4] in 2019.

**Baselines.** We compare our method against the following deep learning-based XMLC models and hierarchical label completion methods :

- **XML-CNN** [15] uses a convolutional neural network with dynamic pooling to learn representations of input documents and to project them onto the output label space.

[6]https://eur-lex.europa.eu/
[7]EuroVoc is EU's multilingual and multidisciplinary thesaurus. It contains keywords, organized in 21 domains and 127 sub-domains in a hierarchical manner https://publications.europa.eu/en/web/eu-vocabularies

- **AttentionXML** [34] first builds a shallow probabilistic label tree (PLT) to partition labels, and then for each level of the constructed PLT trains a deep learning model with multi-label attention.
- **MATCH** [37] leverages documents metadata and a label hierarchy for extreme multi-label classification.
- **XR-Transformer** [35] is a transformer based framework where the pre-trained transformer is recursively fine-tuned on a series of easy-to-hard training objectives defined by a hierarchical label tree.
- **REASSIGN** [25] is a state-of-the-art multi-label completion method, which leverages label hierarchy to aggregate probabilities of individual labels along paths in the tree and select paths with highest aggregated scores. Contrary to matrix completion based techniques, REASSIGN is able to scale to XMLCo problems.

**Implementation and Hyperparameters.** All baselines are re-trained from scratch on our completed versions of the three datasets. We use GloVe.840B.300d as initialized word emdeddings for all models. For baselines, we directly use the default hyperparameter values as provided by the authors. REASSIGN requires a pre-trained classifier to compute the probability of every instance-label association. As such, we trained a vanilla Transformer, i.e. the Transformer encoder for input document representation followed by a fully connected layer to perform multi-label classification. Our model HECTOR was trained using the Adam optimizer with an initial learning rate of 1e-4 and a weight decay of 0.01.

**Metrics.** In line with previous XMLC works [15, 34, 37], we use $P@k$ (Precision at $k$) and $NDCG@k$ (Normalized Discounted Cumulative Gain at $k$) as our evaluation metrics for performance comparison (hereinafter *ranking metrics*). $P@k$ is defined as the number of correct predictions considering only the top $k$ elements divided by $k$:

$$P@k = \frac{1}{k} \sum_{l=1}^{k} y_{rank(l)} \qquad (1)$$

where $y \in \{0, 1\}^L$ is the vector of true labels, and $rank(l)$ is the index of the $l$-th top predicted label. Discounted cumulative gain (DCG) measures the quality of ranking, assigning higher scores to hits at top ranks. nDCG is a normalized version of DCG, which accounts for the varying number of positive labels per instance. $nDCG@k$ is defined by the following formulas:

$$DCG@k = \sum_{l=1}^{k} \frac{y_{rank(l)}}{log(l+1)} \qquad (2)$$

$$nDCG@k = \frac{DCG@k}{\sum_{l=1}^{min(k,||y||_0)} \frac{y_{rank(l)}}{log(l+1)}} \qquad (3)$$

where $||y||_0$ is the number of positive labels in the true label $y$.

To get additional insight about models' performance on low-resource (i.e., corresponding to lower levels of taxonomy) classes, we also report results on *micro_f1* and *macro_f1* (hereinafter *classification metrics*). *micro_f1* is calculated globally by counting the total true positives, false negatives and false positives. For

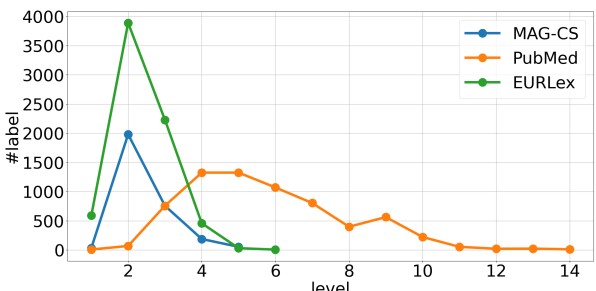

**Figure 4: The number of labels per level of ontology**

*macro_f1*, the metric is calculated for each label, and then their unweighted mean is computed.

## 4.2 Label Refinement

**Experimental design.** For the task of label refinement, each document is accompanied with a set of general labels pertaining to it, and the model must predict more specific labels. In our context, general labels are the labels that belong to the higher levels of the taxonomy, while specific labels are labels of deeper levels. In this set of experiments, we view the label refinement task as a function of $L$, where $L$ is the level from which we start the refinement process. For example, when $L = 3$, we assume that a document is labeled with labels of level 1 and 2 and the task is to predict labels starting from level 3 and deeper. For the baseline methods, we run a normal inference step and then skip model predictions of labels from level 1 to $L - 1$, since we assume that all relevant labels of these levels are provided. Thus we measure the performance on labels of level $L$ and deeper. For ranking metrics, we further rank labels by their predicted probabilities. For classification metrics, we select the best decision boundary for each model and for each experiment. For HECTOR, we use labels from level 1 to $L - 1$ associated with a document as path prefixes and pass them as input to the decoder. More specifically, we build path prefixes from the provided labels, pass them to the decoder as a leftward context and predict the next label(s) in the path starting from the given prefix. All predictions are then merged into a flat list and sorted by their individual scores (see Section 3.3 for more details).

**Results.** We report the key results of our label refinement experiment in Table 2. Interestingly, even when provided with only very general labels (i.e. labels from the first level of taxonomy), HECTOR already significantly outperforms the competing methods across all datasets -– from 2.5% on MAG-CS to 5.9% on EURLex (measured by $P@1$). Furthermore, the advantage of HECTOR on $P@1$ tends to be even more pronounced for higher values of $L$, such as 12% for $L = 3$ on EURLex. Importantly, while this advantage varies with the dataset and the structure of the taxonomy, it is present across all metrics. Notably, AttentionXML performs consistently good across all datasets and is often a close second to HECTOR by ranking metrics on MAG-CS and PubMed datasets. This shows that AttentionXML is a strong baseline for label completion, especially for scientific document collections. Similarly, Transformer-XR, performs well in our experiments, closely following AttentionXML

**Table 2: Performance comparison of HECTOR and other competing methods on Label Refinement task.** $L$ denotes the level of taxonomy, from which the refinement starts. $P@k$ – Precision@k; $N@k$ – nDCG@k.

| L | Algorithms | MAG-CS | | | | PubMed | | | | EURLex | | | |
|---|---|---|---|---|---|---|---|---|---|---|---|---|---|
| | | P@1 | P@3 | N@3 | N@5 | P@1 | P@3 | N@3 | N@5 | P@1 | P@3 | N@3 | N@5 |
| 2 | XML-CNN | 0.7002 | 0.4516 | 0.6366 | 0.6390 | 0.9190 | 0.8942 | 0.9026 | 0.8902 | 0.8998 | 0.8136 | 0.8471 | 0.8147 |
| | AttentionXML | 0.8665 | 0.5884 | 0.8381 | 0.8406 | 0.9288 | 0.9103 | 0.9175 | 0.9082 | 0.9205 | 0.8344 | 0.8676 | 0.8334 |
| | MATCH | 0.8434 | 0.5363 | 0.7795 | 0.7721 | 0.9190 | 0.8967 | 0.9047 | 0.8937 | - | - | - | - |
| | XR-Transformer | 0.8027 | 0.5437 | 0.7677 | 0.7717 | 0.9180 | 0.9041 | 0.9104 | 0.9029 | 0.9276 | 0.8587 | 0.8890 | 0.8568 |
| | REASSIGN | 0.6680 | 0.4224 | 0.5942 | 0.5901 | 0.9196 | 0.8554 | 0.8713 | 0.8417 | 0.8655 | 0.773 | 0.8061 | 0.7691 |
| | HECTOR | **0.8917** | **0.5931** | **0.8530** | **0.8527** | **0.9753** | **0.9436** | **0.9554** | **0.9392** | **0.9861** | **0.9419** | **0.9691** | **0.9563** |
| 3 | XML-CNN | 0.6747 | 0.4121 | 0.6681 | 0.6913 | 0.8993 | 0.8638 | 0.8775 | 0.8681 | 0.8028 | 0.5038 | 0.7942 | 0.8146 |
| | AttentionXML | 0.8346 | 0.4973 | 0.8290 | 0.8448 | 0.9177 | 0.887 | 0.9006 | 0.8925 | 0.8220 | 0.5158 | 0.8111 | 0.8345 |
| | MATCH | 0.7818 | 0.4496 | 0.7583 | 0.7725 | 0.9025 | 0.8691 | 0.8827 | 0.8737 | - | - | - | - |
| | XR-Transformer | 0.7906 | 0.4770 | 0.7879 | 0.8015 | 0.9093 | 0.8827 | 0.8960 | 0.8892 | 0.8441 | 0.5211 | 0.8239 | 0.8343 |
| | REASSIGN | 0.6019 | 0.3636 | 0.5836 | 0.6025 | 0.8916 | 0.8301 | 0.8484 | 0.8238 | 0.7598 | 0.4791 | 0.7522 | 0.7735 |
| | HECTOR | **0.8818** | **0.5141** | **0.8745** | **0.8885** | **0.9754** | **0.9363** | **0.9589** | **0.9468** | **0.9579** | **0.6034** | **0.9506** | **0.9595** |
| 4 | XML-CNN | 0.6662 | 0.3777 | 0.7358 | 0.7724 | 0.8743 | 0.8547 | 0.8650 | 0.8571 | 0.8115 | 0.3690 | 0.8655 | 0.8794 |
| | AttentionXML | 0.8113 | 0.4257 | 0.8581 | 0.8788 | 0.9021 | 0.8816 | 0.8944 | 0.8884 | 0.8251 | 0.3775 | 0.8836 | 0.8957 |
| | MATCH | 0.7330 | 0.3843 | 0.7789 | 0.8071 | 0.8820 | 0.8627 | 0.8747 | 0.8678 | - | - | - | - |
| | XR-Transformer | 0.7775 | 0.4083 | 0.8197 | 0.8364 | 0.8980 | 0.8765 | 0.8907 | 0.8846 | 0.8163 | 0.3448 | 0.8289 | 0.8360 |
| | REASSIGN | 0.5416 | 0.3174 | 0.6015 | 0.6478 | 0.8716 | 0.8469 | 0.8584 | 0.8476 | 0.7636 | 0.3613 | 0.8359 | 0.8518 |
| | HECTOR | **0.8494** | **0.4390** | **0.8961** | **0.9140** | **0.9711** | **0.9294** | **0.9601** | **0.9523** | **0.9177** | **0.3991** | **0.9542** | **0.9583** |
| 5 | XML-CNN | 0.7815 | 0.3376 | 0.8581 | 0.8736 | 0.8926 | 0.8742 | 0.8871 | 0.8742 | 0.9640 | 0.3393 | 0.9739 | 0.9774 |
| | AttentionXML | 0.8612 | 0.3492 | 0.9101 | 0.9209 | 0.9203 | 0.8975 | 0.9150 | 0.9072 | 0.9640 | **0.3483** | 0.9841 | 0.9841 |
| | MATCH | 0.7802 | 0.3256 | 0.8368 | 0.8585 | 0.9026 | 0.8788 | 0.8962 | 0.8877 | - | - | - | - |
| | XR-Transformer | 0.8213 | 0.3243 | 0.8551 | 0.8664 | 0.9139 | 0.8891 | 0.9077 | 0.8997 | 0.9189 | 0.3273 | 0.9346 | 0.9480 |
| | REASSIGN | 0.7121 | 0.3205 | 0.8022 | 0.8283 | 0.8912 | 0.8723 | 0.8857 | 0.8759 | 0.9279 | 0.3393 | 0.9611 | 0.9659 |
| | HECTOR | **0.8946** | **0.3526** | **0.9292** | **0.9370** | **0.9788** | **0.9359** | **0.9711** | **0.9610** | **0.9989** | **0.3483** | **0.9978** | **0.9978** |

**Table 3: Performance Comparison of ablation versions of HECTOR on Label Refinement task with $L = 1$.**

| Dataset | Algorithms | P@1 | N@3 | N@5 |
|---|---|---|---|---|
| MAG-CS | 300_300 | 0.8881 | 0.8247 | 0.8170 |
| | UniSmooth | 0.8813 | 0.8263 | 0.8219 |
| | HECTOR | 0.8918 | 0.8341 | 0.8286 |
| PubMed | 300_300 | 0.9244 | 0.9068 | 0.8890 |
| | UniSmooth | 0.9193 | 0.9001 | 0.8912 |
| | HECTOR | 0.9340 | 0.9173 | 0.9002 |
| PubMed | 300_300 | 0.9207 | 0.8954 | 0.8710 |
| | UniSmooth | 0.9173 | 0.8951 | 0.8779 |
| | HECTOR | 0.9233 | 0.9048 | 0.8809 |

**300_300: HECTOR with 300d label embeddings; UniSmooth: HECTOR with uniform smoothing among all labels.**

in most of the experiments and outperforming it on some. Conversely, XML-CNN tends to perform significantly worse than the other approaches in our experiments. Since XML-CNN is one of the first deep-learning methods for XMLC, it neither features attention mechanism nor transformer architectures, contrary to the other methods considered in our experiments. This further highlights the

advantage of the transformer approach for XMLCo. While MATCH yields the best results for the XML Classification task on MAG-CS dataset [37], its performance turns out to rather low on label refinement tasks. Finally, REASSIGN's performance is subpar in our experiments. This may be explained by the fact that while compatible with XMLCo, REASSIGN is designed for a leaf-mandatory problem, and tends to focus on full paths prediction, resulting in increase weights for labels that are at the deepest level. However, in the different dataset considered in this experiment, many texts are only equipped with labels that are of average depth, and do not include any terminal label, which might considerably deteriorate the performance of the method.

We also report classification metrics ($micro\_f1$ and $macro\_f1$) in Figure 5. Overall, these metrics strengthen our previous observations. HECTOR demonstrates the best, or close to the best, results for both metrics, highlighting its advantage when predicting low-resource classes. For instance, on EURLex with $L = 2$, HECTOR outperforms the next best competing method by 10.9% and 19.4% at $micro\_f1$ and $macro\_f1$, respectively, while on MAG-CS it lags behind by resp. 1.1% and 3.5%. This difference be explained by the properties of the labels tree of each document, that significantly differ across datasets. Indeed, in MAG-CS, label trees are wider, and consequently, documents are tagged with multiple sibling labels, while trees in PubMed and EURLex tend to be narrower, which

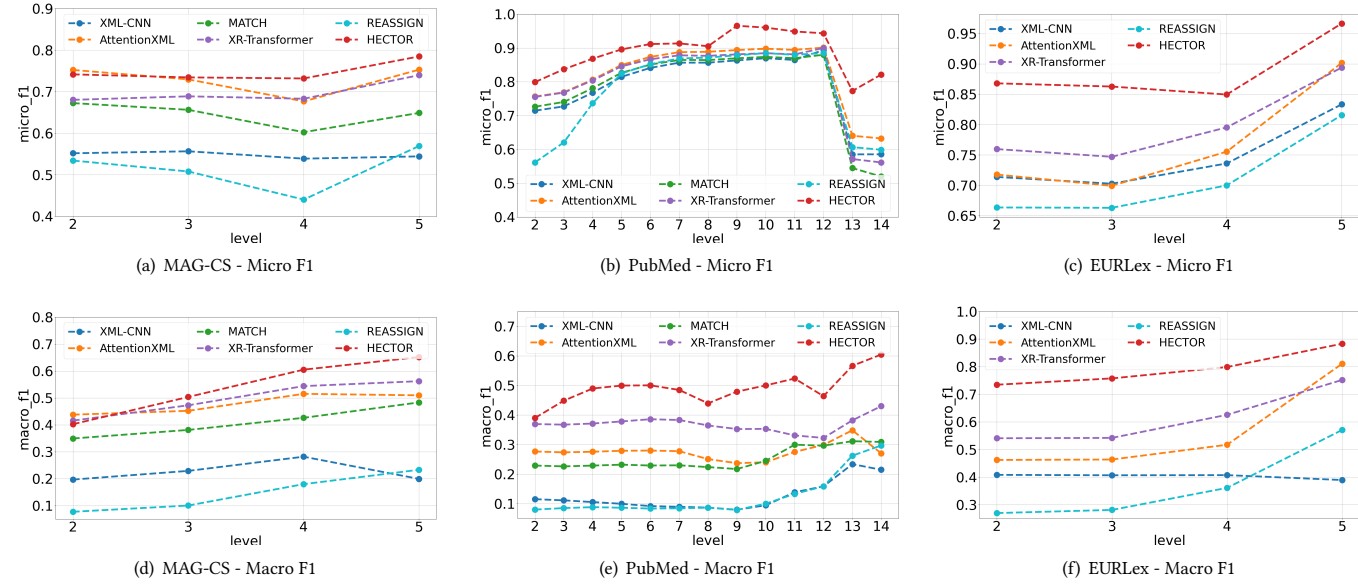

(a) MAG-CS - Micro F1          (b) PubMed - Micro F1          (c) EURLex - Micro F1

(d) MAG-CS - Macro F1          (e) PubMed - Macro F1          (f) EURLex - Macro F1

**Figure 5: Performance comparison of HECTOR and other competing methods on Label Refinement task by Micro F1 and Macro F1 scores. The x-axis represents the taxonomy level $L$ from which we start the refinement process.**

allows to fully leverage HECTOR's sequential path decoding algorithm. Finally, while most baselines perform similarly for classification metrics than for ranking metrics, XR-transformer performance is better for the *micro_f1* and *macro_f1*, where it mostly outperforms other baselines. This demonstrates that XR-Transformer has a consistent performance across all labels, including low-resource ones. In summary, these results highlight the effectiveness of HECTOR for both XMLCo and classification metrics on the label refinement task. For completion purpose, we report further metrics in the supplementary material that further illustrate this observation.

### 4.3 Ablation Study

Finally, we perform ablation studies to justify specific design choices discussed in Section 3.3. In particular, we aim at evaluating the impact of the 600d label embeddings and the smoothing loss function. The results of this experiment are reported in Table 3.

**600d label embeddings.** HECTOR uses 300d GloVe embeddings as initial word representation, and 600d embeddings for label representation to ensure better separability in the vector space. To evaluate the impact of 600d label embeddings, we trained an ablation version of the full HECTOR model where both word and label embeddings are 300-dimensional – HECTOR 300_300. The adapter between the encoder and the decoder is eliminated in this architecture, since there is no need for dimension expansion. The results of this experiment are reported in Table 3. HECTOR 300_300 perform slightly worse than HECTOR on all three datasets, justifying the choice of a 600 dimensional embedding and of the adapter.

**Smoothing by level.** The loss function used for the training of HECTOR incorporates prior knowledge about label taxonomy into the smoothing function: at each position in the output sequence (each level of the label tree), the smoothing value is uniformly

distributed among the labels of the corresponding level rather than all available labels. To evaluate our smoothing-by-level algorithm, we trained HECTOR UniSmooth – a variation of HECTOR with a smoothing value uniformly distributed among *all* false labels. This way, the model does not know in advance which labels are valid at a specific position and learns the taxonomy structure from data alone. Experimental results reported in Table 3 indicate that incorporating prior knowledge about the taxonomy into the model improves model performance. The improvement is especially evident at $P@1$, which corresponds to the prediction of the first label in a path. This can be explained by the fact that at the start of the path there is no left context and the task of predicting the first label is particularly challenging for the decoder, hence it profits from a reduced search space. We also note that although HECTOR UniSmooth performs worse than the full HECTOR model, it still demonstrates strong performance, which shows that our method is capable of learning the structure of labels without any prior knowledge.

## 5 CONCLUSION AND FUTURE WORK

In this paper, we introduced a novel paradigm in the context of XMLCo, where labels are predicted as paths on a hierarchical label tree. This paradigm allowed us to reformulate XMLCo as a Seq2Seq task. Our proposed approach, HECTOR, is able to leverage the transformer architecture on this task to model fine-grained dependencies between text tokens and labels and encode meta-information contained in hierarchical label trees, resulting in HECTOR substantially outperforms state-of-the-art methods on label refinement tasks in our experiments, across all considered datasets. Future works include the use of different ensemble techniques in combination with Hector to further improve our results.

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

# A   ADDITIONAL EXPERIMENTAL RESULTS

## A.1   Label Refinement

In this section we report the full results on our label refinement experiments (Table 4, Table 5 and Table 6 for MAG-CS, PubMed and EURLex datasets, respectively). These tables contain additional metrics, namely micro and macro Precision and Recall, as well as several additional values of ranking metrics. Overall the new metrics confirm the observations and conclusions made in Section 3.4. We refer the reader to Section 4 of the main paper for the complete discussion around our experimental design and results.

## A.2   Extreme Multi-Label Classification

We additionally evaluate HECTOR on a traditional XMLC task, i.e., with no labels provided. We note that this is *not* the task for which HECTOR was developed and that this is not the focus of the present work. Despite this, our method still demonstrates competitive performance compared to other state-of-the-art methods and outperforms several of the baselines in this different context also. We report results on the XMLC task in Table 7.

**Table 4: Performance comparison of HECTOR and other competing methods on Label Refinement task on MAG-CS dataset.** $L$ denotes the level of taxonomy, from which the refinement starts. $P@k$ – **Precision@k**; $N@k$ – **nDCG@k**; $\mu X$ – **micro average**; $MX$ – **macro average**.

| L | Algorithms | $P@1$ | $P@3$ | $P@5$ | $N@3$ | $N@5$ | $\mu Prec$ | $\mu Recall$ | $\mu F1$ | $MPrec$ | $MRecall$ | $MF1$ |
|---|---|---|---|---|---|---|---|---|---|---|---|---|
| | XML-CNN | 0.7002 | 0.4516 | 0.3283 | 0.6366 | 0.6390 | 0.6658 | 0.4712 | 0.5518 | 0.2639 | 0.1570 | 0.1969 |
| | AttentionXML | 0.8665 | 0.5884 | **0.4245** | 0.8381 | 0.8406 | **0.7965** | **0.7125** | **0.7522** | **0.4073** | 0.4732 | **0.4378** |
| 2 | MATCH | 0.8434 | 0.5363 | 0.3763 | 0.7795 | 0.7721 | 0.7826 | 0.5895 | 0.6724 | 0.3989 | 0.3107 | 0.3494 |
| | XR-Transformer | 0.8027 | 0.5437 | 0.3958 | 0.7677 | 0.7717 | 0.7325 | 0.6350 | 0.6803 | 0.3926 | 0.4417 | 0.4157 |
| | REASSIGN | 0.6680 | 0.4224 | 0.3017 | 0.5942 | 0.5901 | 0.7054 | 0.4300 | 0.5343 | 0.0709 | 0.0851 | 0.0773 |
| | HECTOR | **0.8917** | **0.5931** | 0.4239 | **0.8530** | **0.8527** | 0.7745 | 0.7113 | 0.7416 | 0.3397 | **0.4936** | 0.4025 |
| | XML-CNN | 0.6747 | 0.4121 | 0.2931 | 0.6681 | 0.6913 | 0.6115 | 0.5106 | 0.5565 | 0.3213 | 0.1781 | 0.2291 |
| | AttentionXML | 0.8346 | 0.4973 | 0.3440 | 0.8290 | 0.8448 | **0.8042** | 0.6674 | 0.7294 | 0.4362 | 0.4693 | 0.4521 |
| 3 | MATCH | 0.7818 | 0.4496 | 0.3097 | 0.7583 | 0.7725 | 0.7381 | 0.5909 | 0.6563 | 0.3979 | 0.3658 | 0.3812 |
| | XR-Transformer | 0.7906 | 0.4770 | 0.3297 | 0.7879 | 0.8015 | 0.7432 | 0.6417 | 0.6887 | **0.4540** | 0.4927 | 0.4726 |
| | REASSIGN | 0.6019 | 0.3636 | 0.2574 | 0.5836 | 0.6025 | 0.6879 | 0.4027 | 0.5080 | 0.0846 | 0.1241 | 0.1006 |
| | HECTOR | **0.8818** | **0.5141** | **0.3521** | **0.8745** | **0.8885** | 0.7513 | **0.7181** | **0.7343** | 0.4106 | **0.6507** | **0.5035** |
| | XML-CNN | 0.6662 | 0.3777 | 0.2555 | 0.7358 | 0.7724 | 0.5899 | 0.4959 | 0.5388 | 0.4167 | 0.2129 | 0.2818 |
| | AttentionXML | 0.8113 | 0.4257 | 0.2748 | 0.8581 | 0.8788 | **0.7311** | 0.6297 | 0.6766 | 0.4691 | 0.5711 | 0.5151 |
| 4 | MATCH | 0.7330 | 0.3843 | 0.2547 | 0.7789 | 0.8071 | 0.6675 | 0.5491 | 0.6025 | 0.3876 | 0.4731 | 0.4261 |
| | XR-Transformer | 0.7775 | 0.4083 | 0.2607 | 0.8197 | 0.8364 | 0.7053 | 0.6620 | 0.6830 | **0.5378** | 0.5499 | 0.5438 |
| | REASSIGN | 0.5416 | 0.3174 | 0.2250 | 0.6015 | 0.6478 | 0.4013 | 0.4879 | 0.4404 | 0.1341 | 0.2735 | 0.1799 |
| | HECTOR | **0.8494** | **0.4390** | **0.2814** | **0.8961** | **0.9140** | 0.7084 | **0.7567** | **0.7317** | 0.5217 | **0.7197** | **0.6049** |
| | XML-CNN | 0.7815 | 0.3376 | 0.2126 | 0.8581 | 0.8736 | **0.8454** | 0.4014 | 0.5443 | 0.3845 | 0.1346 | 0.1994 |
| | AttentionXML | 0.8612 | 0.3492 | 0.2162 | 0.9101 | 0.9209 | 0.7526 | 0.7534 | 0.7530 | 0.5181 | 0.5013 | 0.5096 |
| 5 | MATCH | 0.7802 | 0.3256 | 0.2080 | 0.8368 | 0.8585 | 0.8370 | 0.5298 | 0.6489 | **0.5886** | 0.4099 | 0.4832 |
| | XR-Transformer | 0.8213 | 0.3243 | 0.2015 | 0.8551 | 0.8664 | 0.7735 | 0.7087 | 0.7397 | 0.5512 | 0.5733 | 0.5621 |
| | REASSIGN | 0.7121 | 0.3205 | 0.2067 | 0.8022 | 0.8283 | 0.5365 | 0.6067 | 0.5694 | 0.2213 | 0.2466 | 0.2333 |
| | HECTOR | **0.8946** | **0.3526** | **0.2170** | **0.9292** | **0.9370** | 0.7675 | **0.8028** | **0.7848** | 0.5704 | **0.7597** | **0.6516** |

Table 5: Performance comparison of HECTOR and other competing methods on Label Refinement task on PubMed dataset. $L$ denotes the level of taxonomy, from which the refinement starts. $P@k$ – Precision@k; $N@k$ – nDCG@k; $\mu X$ – micro average; $MX$ – macro average.

| L | Algorithms | P@1 | P@3 | P@5 | N@3 | N@5 | μPrec | μRecall | μF1 | MPrec | MRecall | MF1 |
|---|---|---|---|---|---|---|---|---|---|---|---|---|
| 2 | XML-CNN | 0.9190 | 0.8942 | 0.8723 | 0.9026 | 0.8902 | 0.8303 | 0.6271 | 0.7145 | 0.1007 | 0.1350 | 0.1153 |
| | AttentionXML | 0.9288 | 0.9103 | 0.8914 | 0.9175 | 0.9082 | 0.8003 | 0.7170 | 0.7563 | 0.2583 | 0.2974 | 0.2765 |
| | MATCH | 0.9190 | 0.8967 | 0.8759 | 0.9047 | 0.8937 | 0.8114 | 0.6571 | 0.7261 | 0.2162 | 0.2434 | 0.2290 |
| | XR-Transformer | 0.9180 | 0.9041 | 0.8867 | 0.9104 | 0.9029 | 0.8176 | 0.7016 | 0.7552 | 0.3720 | 0.3667 | 0.3693 |
| | REASSIGN | 0.9196 | 0.8554 | 0.8132 | 0.8713 | 0.8417 | **0.8672** | 0.4151 | 0.5615 | 0.0525 | 0.1675 | 0.0800 |
| | HECTOR | **0.9753** | **0.9436** | **0.9101** | **0.9554** | **0.9392** | 0.7967 | **0.8011** | 0.7989 | 0.3758 | **0.4051** | **0.3899** |
| 3 | XML-CNN | 0.8993 | 0.8638 | 0.8443 | 0.8775 | 0.8681 | 0.8488 | 0.6360 | 0.7271 | 0.0984 | 0.1282 | 0.1114 |
| | AttentionXML | 0.9177 | 0.8870 | 0.8674 | 0.9006 | 0.8925 | 0.8171 | 0.7262 | 0.7690 | 0.2575 | 0.2920 | 0.2737 |
| | MATCH | 0.9025 | 0.8691 | 0.8487 | 0.8827 | 0.8737 | 0.8342 | 0.6653 | 0.7402 | 0.2147 | 0.2390 | 0.2262 |
| | XR-Transformer | 0.9093 | 0.8827 | 0.8636 | 0.8960 | 0.8892 | 0.8346 | 0.7110 | 0.7678 | 0.3708 | 0.3637 | 0.3673 |
| | REASSIGN | 0.8916 | 0.8301 | 0.7931 | 0.8484 | 0.8238 | **0.8927** | 0.4757 | 0.6207 | 0.0614 | 0.1391 | 0.0852 |
| | HECTOR | **0.9754** | **0.9363** | **0.9019** | **0.9589** | **0.9468** | 0.8295 | **0.8445** | 0.8369 | 0.4173 | **0.4842** | **0.4483** |
| 4 | XML-CNN | 0.8743 | 0.8547 | 0.8334 | 0.8650 | 0.8571 | **0.8734** | 0.6844 | 0.7674 | 0.0957 | 0.1176 | 0.1055 |
| | AttentionXML | 0.9021 | 0.8816 | 0.8597 | 0.8944 | 0.8884 | 0.8491 | 0.7678 | 0.8064 | 0.2569 | 0.2974 | 0.2757 |
| | MATCH | 0.8820 | 0.8627 | 0.8401 | 0.8747 | 0.8678 | 0.8666 | 0.7107 | 0.7809 | 0.2209 | 0.2375 | 0.2289 |
| | XR-Transformer | 0.8980 | 0.8765 | 0.8538 | 0.8907 | 0.8846 | 0.8384 | 0.7723 | 0.8040 | 0.3762 | 0.3661 | 0.3711 |
| | REASSIGN | 0.8716 | 0.8469 | 0.8225 | 0.8584 | 0.8476 | 0.7802 | 0.6978 | 0.7367 | 0.0684 | 0.1249 | 0.0884 |
| | HECTOR | **0.9711** | **0.9294** | **0.8937** | **0.9601** | **0.9523** | 0.8561 | **0.8808** | 0.8683 | 0.4587 | **0.5248** | **0.4895** |
| 5 | XML-CNN | 0.8926 | 0.8742 | 0.8447 | 0.8871 | 0.8742 | **0.8961** | 0.7467 | 0.8146 | 0.0822 | 0.1268 | 0.0998 |
| | AttentionXML | 0.9203 | 0.8975 | 0.8709 | 0.9150 | 0.9072 | 0.8836 | 0.8183 | 0.8497 | 0.2526 | 0.3114 | 0.2789 |
| | MATCH | 0.9026 | 0.8788 | 0.8520 | 0.8962 | 0.8877 | 0.8959 | 0.7672 | 0.8266 | 0.2261 | 0.2382 | 0.2320 |
| | XR-Transformer | 0.9139 | 0.8891 | 0.8622 | 0.9077 | 0.8997 | 0.8732 | 0.8197 | 0.8456 | 0.3854 | 0.3712 | 0.3781 |
| | REASSIGN | 0.8912 | 0.8723 | 0.8467 | 0.8857 | 0.8759 | 0.8930 | 0.7656 | 0.8244 | 0.0675 | 0.1199 | 0.0863 |
| | HECTOR | **0.9788** | **0.9359** | **0.8956** | **0.9711** | **0.9610** | 0.8870 | **0.9047** | 0.8958 | 0.4564 | **0.5514** | **0.4994** |
| 6 | XML-CNN | 0.9094 | 0.8801 | 0.8564 | 0.8980 | 0.8879 | 0.9035 | 0.7866 | 0.8410 | 0.0874 | 0.0969 | 0.0919 |
| | AttentionXML | 0.9368 | 0.9049 | 0.8815 | 0.9276 | 0.9213 | 0.9012 | 0.8475 | 0.8735 | 0.2632 | 0.2985 | 0.2797 |
| | MATCH | 0.9170 | 0.8855 | 0.8614 | 0.9073 | 0.9002 | 0.8983 | 0.8082 | 0.8509 | 0.2240 | 0.2346 | 0.2291 |
| | XR-Transformer | 0.9268 | 0.8941 | 0.8700 | 0.9172 | 0.9102 | 0.8867 | 0.8472 | 0.8665 | 0.3933 | 0.3784 | 0.3857 |
| | REASSIGN | 0.9087 | 0.8818 | 0.8606 | 0.8994 | 0.8924 | 0.9046 | 0.8055 | 0.8522 | 0.0673 | 0.1102 | 0.0836 |
| | HECTOR | **0.9832** | **0.9341** | **0.8999** | **0.9730** | **0.9648** | **0.9053** | **0.9178** | 0.9115 | 0.4546 | **0.5561** | **0.5003** |
| 7 | XML-CNN | 0.9429 | 0.8984 | 0.8790 | 0.9123 | 0.9050 | 0.9076 | 0.8108 | 0.8565 | 0.0712 | 0.1204 | 0.0895 |
| | AttentionXML | 0.9583 | 0.9241 | 0.9031 | 0.9396 | 0.9351 | **0.9140** | 0.8625 | 0.8875 | 0.2630 | 0.2939 | 0.2776 |
| | MATCH | 0.9392 | 0.9043 | 0.8828 | 0.9197 | 0.9147 | 0.9071 | 0.8256 | 0.8644 | 0.1911 | 0.2883 | 0.2298 |
| | XR-Transformer | 0.9447 | 0.9108 | 0.8888 | 0.9262 | 0.9207 | 0.8951 | 0.8620 | 0.8782 | 0.3976 | 0.3694 | 0.3830 |
| | REASSIGN | 0.9403 | 0.9043 | 0.8869 | 0.9167 | 0.9122 | 0.9118 | 0.8299 | 0.8689 | 0.0648 | 0.1211 | 0.0844 |
| | HECTOR | **0.9873** | **0.9430** | **0.9143** | **0.9682** | **0.9617** | 0.9007 | **0.9263** | 0.9133 | 0.4356 | **0.5459** | **0.4846** |
| 8 | XML-CNN | 0.9066 | 0.8973 | 0.8706 | 0.9058 | 0.9020 | 0.9007 | 0.8164 | 0.8565 | 0.0726 | 0.1079 | 0.0868 |
| | AttentionXML | 0.9383 | 0.9230 | 0.8934 | 0.9375 | 0.9353 | **0.9143** | 0.8642 | 0.8885 | 0.2388 | 0.2639 | 0.2507 |
| | MATCH | 0.9171 | 0.9016 | 0.8701 | 0.9160 | 0.9122 | 0.9036 | 0.8280 | 0.8641 | 0.1872 | 0.2782 | 0.2238 |
| | XR-Transformer | 0.9234 | 0.9070 | 0.8766 | 0.9219 | 0.9184 | 0.8901 | 0.8636 | 0.8767 | 0.3854 | 0.3458 | 0.3645 |
| | REASSIGN | 0.9167 | 0.9054 | 0.8781 | 0.9153 | 0.9132 | 0.9079 | 0.8379 | 0.8715 | 0.0605 | 0.1521 | 0.0866 |
| | HECTOR | **0.9593** | **0.9348** | **0.9005** | **0.9557** | **0.9524** | 0.8839 | **0.9273** | 0.9051 | 0.4053 | **0.4797** | **0.4394** |
| 9 | XML-CNN | 0.9136 | 0.8975 | 0.7334 | 0.9111 | 0.9137 | 0.9038 | 0.8260 | 0.8631 | 0.0654 | 0.1019 | 0.0797 |
| | AttentionXML | 0.9436 | 0.9211 | 0.7501 | 0.9421 | 0.9437 | 0.9197 | 0.8692 | 0.8937 | 0.1963 | 0.2993 | 0.2371 |
| | MATCH | 0.9228 | 0.9001 | 0.7281 | 0.9209 | 0.9194 | 0.9088 | 0.8333 | 0.8694 | 0.1880 | 0.2573 | 0.2172 |
| | XR-Transformer | 0.9276 | 0.9041 | 0.7319 | 0.9255 | 0.9236 | 0.8936 | 0.8689 | 0.8810 | 0.3375 | 0.3688 | 0.3524 |
| | REASSIGN | 0.9229 | 0.9084 | 0.7384 | 0.9237 | 0.9245 | 0.9122 | 0.8470 | 0.8784 | 0.0551 | 0.1387 | 0.0789 |
| | HECTOR | **0.9926** | **0.9621** | **0.7782** | **0.9902** | **0.9893** | **0.9565** | **0.9748** | **0.9655** | 0.3904 | **0.6192** | **0.4789** |
| 10 | XML-CNN | 0.9370 | 0.8981 | 0.6010 | 0.9287 | 0.9432 | 0.9107 | 0.8321 | 0.8696 | 0.0747 | 0.1292 | 0.0947 |
| | AttentionXML | 0.9539 | 0.9156 | 0.6095 | 0.9495 | 0.9605 | 0.9258 | 0.8710 | 0.8976 | 0.1929 | 0.3181 | 0.2401 |
| | MATCH | 0.9340 | 0.8924 | 0.5895 | 0.9271 | 0.9351 | 0.9041 | 0.8449 | 0.8735 | 0.2248 | 0.2693 | 0.2451 |
| | XR-Transformer | 0.9350 | 0.8969 | 0.5899 | 0.9301 | 0.9355 | 0.8987 | 0.8708 | 0.8846 | 0.3450 | 0.3614 | 0.3530 |
| | REASSIGN | 0.9431 | 0.9053 | 0.6032 | 0.9387 | 0.9504 | 0.9180 | 0.8536 | 0.8846 | 0.0686 | 0.1854 | 0.1002 |
| | HECTOR | **0.9957** | **0.9474** | **0.6223** | **0.9900** | **0.9914** | **0.9470** | **0.9738** | **0.9602** | 0.4263 | **0.6046** | **0.5000** |

**Table 6: Performance comparison of HECTOR and other competing methods on Label Refinement task on EURLex dataset.** $L$ denotes the level of taxonomy, from which the refinement starts. $P@k$ – Precision@k; $N@k$ – nDCG@k; $\mu X$ – micro average; $MX$ – macro average.

| L | Algorithms | P@1 | P@3 | P@5 | N@3 | N@5 | μPrec | μRecall | μF1 | MPrec | MRecall | MF1 |
|---|---|---|---|---|---|---|---|---|---|---|---|---|
| 2 | XML-CNN | 0.8998 | 0.8136 | 0.7082 | 0.8471 | 0.8147 | 0.7823 | 0.6560 | 0.7136 | 0.3937 | 0.4236 | 0.4081 |
| | AttentionXML | 0.9205 | 0.8344 | 0.7249 | 0.8676 | 0.8334 | 0.7618 | 0.6782 | 0.7176 | 0.3965 | 0.5537 | 0.4621 |
| | MATCH | - | - | - | - | - | - | - | - | - | - | - |
| | XR-Transformer | 0.9276 | 0.8587 | 0.7486 | 0.8890 | 0.8568 | 0.7924 | 0.7293 | 0.7595 | 0.4869 | 0.6077 | 0.5407 |
| | REASSIGN | 0.8655 | 0.7730 | 0.6668 | 0.8061 | 0.7691 | 0.6758 | 0.6513 | 0.6633 | 0.2120 | 0.3704 | 0.2697 |
| | HECTOR | **0.9861** | **0.9419** | **0.8494** | **0.9691** | **0.9563** | **0.8170** | **0.9258** | **0.8680** | **0.6863** | **0.7902** | **0.7346** |
| 3 | XML-CNN | 0.8028 | 0.5038 | 0.3509 | 0.7942 | 0.8146 | 0.7510 | 0.6598 | 0.7025 | 0.4064 | 0.4069 | 0.4066 |
| | AttentionXML | 0.8220 | 0.5158 | 0.3618 | 0.8111 | 0.8345 | 0.7200 | 0.6792 | 0.6990 | 0.3874 | 0.5773 | 0.4636 |
| | MATCH | - | - | - | - | - | - | - | - | - | - | - |
| | XR-Transformer | 0.8441 | 0.5211 | 0.3551 | 0.8239 | 0.8343 | 0.7600 | 0.7342 | 0.7469 | 0.5037 | 0.5866 | 0.5420 |
| | REASSIGN | 0.7598 | 0.4791 | 0.3350 | 0.7522 | 0.7735 | 0.6944 | 0.6338 | 0.6627 | 0.2265 | 0.3703 | 0.2810 |
| | HECTOR | **0.9579** | **0.6034** | **0.4081** | **0.9506** | **0.9595** | **0.8091** | **0.9239** | **0.8627** | **0.7414** | **0.7744** | **0.7575** |
| 4 | XML-CNN | 0.8115 | 0.3690 | 0.2310 | 0.8655 | 0.8794 | 0.8115 | 0.6731 | 0.7359 | 0.4410 | 0.3781 | 0.4071 |
| | AttentionXML | 0.8251 | 0.3775 | 0.2350 | 0.8836 | 0.8957 | 0.8069 | 0.7099 | 0.7553 | 0.4173 | 0.6799 | 0.5171 |
| | MATCH | - | - | - | - | - | - | - | - | - | - | - |
| | XR-Transformer | 0.8163 | 0.3448 | 0.2125 | 0.8289 | 0.8360 | **0.8393** | 0.7558 | 0.7954 | 0.6044 | 0.6483 | 0.6256 |
| | REASSIGN | 0.7636 | 0.3613 | 0.2276 | 0.8359 | 0.8518 | 0.7152 | 0.6848 | 0.6997 | 0.3027 | 0.4469 | 0.3609 |
| | HECTOR | **0.9177** | **0.3991** | **0.2435** | **0.9542** | **0.9583** | 0.7858 | **0.9244** | **0.8495** | **0.7669** | **0.8334** | **0.7988** |
| 5 | XML-CNN | 0.9640 | 0.3393 | 0.2054 | 0.9739 | 0.9774 | **0.9659** | 0.7328 | 0.8333 | 0.4480 | 0.3442 | 0.3893 |
| | AttentionXML | 0.9640 | **0.3483** | **0.2090** | 0.9841 | 0.9841 | 0.9352 | 0.8707 | 0.9018 | 0.7726 | 0.8534 | 0.8110 |
| | MATCH | - | - | - | - | - | - | - | - | - | - | - |
| | XR-Transformer | 0.9189 | 0.3273 | 0.2036 | 0.9346 | 0.9480 | 0.9604 | 0.8362 | 0.8940 | **0.9021** | 0.6449 | 0.7521 |
| | REASSIGN | 0.9279 | 0.3393 | 0.2072 | 0.9611 | 0.9659 | 0.9333 | 0.7241 | 0.8155 | 0.5504 | 0.5932 | 0.5710 |
| | HECTOR | **0.9989** | **0.3483** | **0.2090** | **0.9978** | **0.9978** | 0.9355 | **0.9974** | **0.9667** | 0.8636 | **0.9037** | **0.8832** |

**Table 7: Performance comparison of HECTOR and other competing methods on the XMLC task.** $P@k$ – Precision@k; $N@k$ – nDCG@k; $\mu X$ – micro average; $MX$ – macro average.

| Dataset | Algorithms | P@1 | P@3 | P@5 | N@3 | N@5 | μPrec | μRecall | μF1 | MPrec | MRecall | MF1 |
|---|---|---|---|---|---|---|---|---|---|---|---|---|
| MAG-CS | XML-CNN | 0.8628 | 0.7049 | 0.5555 | 0.7819 | 0.7638 | 0.6666 | 0.6042 | 0.6338 | 0.2558 | 0.1655 | 0.2010 |
| | AttentionXML | 0.8830 | 0.7732 | **0.6336** | 0.8397 | 0.8395 | 0.7404 | **0.7383** | **0.7394** | **0.3907** | **0.4723** | **0.4276** |
| | MATCH | **0.9228** | **0.7797** | 0.6182 | **0.8574** | **0.8421** | **0.7604** | 0.6814 | 0.7187 | 0.3872 | 0.3168 | 0.3484 |
| | XR-Transformer | 0.8607 | 0.7309 | 0.5886 | 0.8008 | 0.7905 | 0.7244 | 0.6489 | 0.6846 | 0.3692 | 0.4455 | 0.4038 |
| | REASSIGN | 0.8706 | 0.7023 | 0.5512 | 0.7808 | 0.7604 | 0.6555 | 0.6079 | 0.6308 | 0.0730 | 0.0957 | 0.0828 |
| | HECTOR | 0.8918 | 0.7616 | 0.6155 | 0.8341 | 0.8286 | 0.7073 | 0.7016 | 0.7045 | 0.3263 | 0.3663 | 0.3451 |
| PubMed | XML-CNN | 0.9408 | 0.9231 | 0.9007 | 0.9272 | 0.9145 | 0.8123 | 0.6507 | 0.7226 | 0.1012 | 0.1363 | 0.1162 |
| | AttentionXML | 0.9434 | **0.9317** | **0.9132** | **0.9344** | **0.9249** | 0.7931 | **0.7304** | **0.7604** | 0.2587 | 0.2984 | 0.2771 |
| | MATCH | 0.9418 | 0.9231 | 0.9024 | 0.9275 | 0.9161 | 0.8047 | 0.6709 | 0.7317 | 0.2168 | 0.2443 | 0.2297 |
| | XR-Transformer | 0.9401 | 0.9246 | 0.9077 | 0.9281 | 0.9199 | 0.8110 | 0.7130 | 0.7589 | **0.3723** | 0.3674 | **0.3698** |
| | REASSIGN | **0.9446** | 0.9055 | 0.8647 | 0.9154 | 0.8880 | **0.8660** | 0.4374 | 0.5812 | 0.0520 | 0.1701 | 0.0797 |
| | HECTOR | 0.9340 | 0.9119 | 0.8822 | 0.9173 | 0.9002 | 0.6808 | 0.7141 | 0.6971 | 0.3548 | 0.2617 | 0.3012 |
| EURLex | XML-CNN | 0.9258 | 0.8922 | 0.8462 | 0.9019 | 0.8734 | 0.7838 | 0.6807 | 0.7287 | 0.4027 | 0.4408 | 0.4209 |
| | AttentionXML | 0.9382 | 0.9083 | 0.8623 | 0.9172 | 0.8887 | 0.7710 | 0.6916 | 0.7291 | 0.4046 | 0.5536 | 0.4675 |
| | MATCH | - | - | - | - | - | - | - | - | - | - | - |
| | XR-Transformer | **0.9417** | **0.9202** | **0.8812** | **0.9270** | **0.9042** | **0.7894** | **0.7466** | **0.7674** | **0.5065** | **0.5894** | **0.5449** |
| | REASSIGN | 0.9162 | 0.8453 | 0.7759 | 0.8629 | 0.8152 | 0.6495 | 0.6247 | 0.6369 | 0.2112 | 0.3581 | 0.2657 |
| | HECTOR | 0.9233 | 0.8972 | 0.8569 | 0.9048 | 0.8809 | 0.7614 | 0.7264 | 0.7435 | 0.4197 | 0.5802 | 0.4871 |

