# OpenReview forum: "Follow the Path: Hierarchy-Aware Extreme Multi-Label Completion for Semantic Text Tagging"
_ACM.org/TheWebConf/2024/Conference — TheWebConf24 Oral_

### Official Review · Reviewer_QAdX · 2023-11-16

**Novelty:** 5
**Technical Quality:** 5

**Review:**

The authors have proposed a Transformer-based approach, named HECTOR, which directly takes advantage of the hierarchical structures of the label space to predict missing labels in multi-label completion tasks. The HECTOR approach performed well as compared to the state-of-the-art methods. The paper is well-written and easy to follow. However, technical novelty is marginal.
Pros:
a) Better results than the baseline methods. HECTOR outperforms the state-of-the-art techniques by more than 10% in different metrics.
b) Uses hierarchical-based technique to solve label refinement.
c) The paper is well written and easy to follow.
Cons:
a) The figures are blurred, e.g. Figure 1 and Figure 2.
b) They have employed a transformer model with a little modification in the input and output, leads to lack in technical novelty in model architecture.
c) Beam search is a state-of-the-art algorithm that plays a vital role in the approach and is not referenced.

**Questions:**

What are the novelties in the model architecture?
In the past, the transformer has been used for predicting missing labels, e.g.XR-Transformer (as mentioned the baseline section). What makes HECTOR work better than XR-Transformer approach?

**Reviewer Confidence:**

4: The reviewer is certain that the evaluation is correct and very familiar with the relevant literature

**Scope:**

4: The work is relevant to the Web and to the track, and is of broad interest to the community

---

### Official Review · Reviewer_xSnn · 2023-11-19

**Novelty:** 6
**Technical Quality:** 5

**Review:**

In this paper, the authors propose a transfORmer-based Text Hierarchical Extreme Completion algorithm to solve the XML Completion problem more efficiently. HECTOR operates by directly predicting the paths in the tree to utilize the information encoded in the hierarchy. Since these paths are sequential, HECTOR can fully utilize the efficiency and performance of the Transformer architecture. The description of the paper is relatively clear and the authors have demonstrated the validity of the method through various experiments. This work skillfully combines the characteristics of the model with the task characteristics and achieves significant performance improvements with good originality.

Pros:
1. HECTOR skillfully combines the model characterization of the Transformer architecture and the characterization of tasks.
2. The paper is relatively clear in its description, rich in experiments, and achieves significant performance improvements.

Cons:
1. The technical details of the problem and model are not defined in sufficiently clear notations.
2. Insufficient exploration of the model architecture, e.g., how well does it perform on some simple data, does the performance mainly benefit from the effectiveness of the transformer on Seq2Seq or the integration of the labeled paths, which may need more ablation or qualitative analysis to be reflected.

**Questions:**

1. Authors should use a more formulaic approach for defining the problem and modeling technical details, which helps improve the clarity of the article and is helpful to the reader.
2. The authors skillfully combine model features with task features, and the proposed approach is illuminating. However, the ablation of the model in this paper does not appear to be complete. While there is some overlap in the contribution of Transformer and label structure information to the model, showing the performance of simpler ablation models would be helpful.
3. The future work section should list more deficiencies and give some guidance.
4. Figure 1 has less clarity.

**Reviewer Confidence:**

3: The reviewer is confident but not certain that the evaluation is correct

**Scope:**

4: The work is relevant to the Web and to the track, and is of broad interest to the community

---

### Official Review · Reviewer_L5SB · 2023-11-22

**Novelty:** 6
**Technical Quality:** 5

**Review:**

The paper proposes a new approach to support extreme multi-label classification (XMLC), and, especially, the case of label refinement. The paper introduces the idea of interpreting XMLC as the problem of decoding label paths based on the encoding of the document and some input labels. The approach is based on an encoder-decoder transformer, which comes with some adaptation, including dimensionality rescaling and injection of prior knowledge via smoothing in the loss function. Experiments on three benchmark datasets show the improvements introduced by the proposed model

PROS

The paper addresses the relevant problem of hierarchical text classification

To the best of my knowledge, the paper introduces quite a novel idea and several mechanisms to implement it

I liked several aspects of the proposed idea, which in my opinion find a clever and interesting way of processing several relevant signals for hierarchical inference (hierarchical information via smoothing, lexical relatedness of class labels via embeddings, etc.)

To the best of my knowledge, the experimental results are quite strong, and do a good job at convincing that the approach works well for label refinement.

CONS

If we consider the call for papers (https://www2024.thewebconf.org/calls/research-tracks/ ) the paper does not comply with requirements, because it fails to discuss the relevance of the proposed approach to the Web context. However, in this particular case, I will not consider this as a criterion for rejection because the relevance of hierarchical text classification for organizing web data is obvious to me. However, the authors should consider the guidelines and discuss this relevance in the introduction.

While the paper is quite well written, there are crucial aspects of the paper that I missed. I felt that too much attention is dedicated to aspects that are quite obvious to me, rather than to technical details that require clarification or at least a better presentation to be understood more easily. This also raises a few questions that must be answered. I have added specific suggestions below.


* Technical comments/requests for clarification

While, to the best of my knowledge, the proposed idea is novel for XMLC, I think that some references to approaches that share some features of the proposed idea should be added: autoregressive entity linking [1] (the proposed approach applies to a multi-label classification problem an idea that resembles that one) and open vocabulary classification (also applied to EurLex) [2]
I am not sure that you need the strong constraint that the taxonomy is a tree; what would prevent your approach from working with a partial order (no tree structure is required)?

Would it be possible to apply the model when the input label is not available, e.g., by starting with a start node? How would the proposed approach and its competitors behave in this setting? I consider label refinement an important real-world problem, but I think it would be fair to discuss these aspects in comparison with the other models. EDIT: I found the answer in the Appendix. I am fine with these results, but I think that this discussion must be included in the main part of the paper (the table could go in the appendix).

Some labels may consist of different tokens; are they handled smoothly because beam search is used to constrain decoding?

I did not find it completely clear how the input to the encoder is shaped. I guess that 300 refers to each token embedding, but I also expect that you have a sequence of tokens (how long is this sequence?). Also, it is not clear why positional embeddings (or positional transformation of input embeddings) are not considered relevant here.


* Presentation comments

Figure 1 and Figure 2 explain topics that are quite understandable and do not add too much value to comprehension. I am not against it, but I think that the space can be used better to illustrate two aspects that are key aspects of the proposed approach and less clear: 1) the smoothing approach used in the loss in lines 434-442; 2) section 3.4 (an example of generated paths and consequent transformation in ranked list of labels would much appreciated), 3) inference for label refinement, i.e., lines 674-679 (an example of input processing would be helpful). Probably, an example that extends Figure 3 with some input for the encoder and the decoder could also be helpful and replace some of the previous suggestions (e.g., 2 and 3)

316 - … I suggest improving this piece of explanation because as a reader I had the impression that your model was defined to predict from fine to coarse labels (which does not make sense).

669 … while reading this explanation I did not understand well if, based on your settings, you remove coarser-grain labels for evaluation, or you consider them but somehow do not focus on them. Also, it is not clear at which level L is fixed, e.g., for an experiment, for a dataset, for individual instances (this is clear later, but I suggest clarifying this earlier).

[1]  De Cao, N., Izacard, G., Riedel, S., & Petroni, F. (2020, September). Autoregressive Entity Retrieval. In ICLR 2021-9th International Conference on Learning Representations (Vol. 2021). ICLR.)

[2] Simig, D., Petroni, F., Yanki, P., Popat, K., Du, C., Riedel, S., & Yazdani, M. (2022, May). Open Vocabulary Extreme Classification Using Generative Models. In Findings of the Association for Computational Linguistics: ACL 2022 (pp. 1561-1583).

**Questions:**

Some labels may consist of different tokens; are they handled smoothly because beam search is used to constrain decoding? And can you clarify how the document input is processed by the encoder?

It is quite surprising that your approach does not outperform a few other methods on full-fledged XMLC without prior label, as I expected that the prediction of first-level labels would have been quite easy. Do you have any explanation for this?

I am not sure that you need the strong constraint that the taxonomy is a tree; what would prevent your approach from working with a partial order (no tree structure is required)?

Will the code be released to ensure replicability?

**Ethics Review Description:**

I do not see any ethics flag for this paper

**Reviewer Confidence:**

3: The reviewer is confident but not certain that the evaluation is correct

**Scope:**

3: The work is somewhat relevant to the Web and to the track, and is of narrow interest to a sub-community

---

### Official Review · Reviewer_wykC · 2023-11-23

**Novelty:** 5
**Technical Quality:** 5

**Review:**

The authors present their research on predicting labels by incorporating taxonomy knowledge into the process.
The research is well motivated and the methodology is well described in detail.

The evaluation was conducted against several baselines, and HECTOR, the proposed approach showed promising results compared to the existing algorithms.

The work is motivated by explaining the importance of discovering semantic tags from text documents to improve content discovery on the web. I would recommend to further reflect on this point to increase the relevance of the submission to the track at TheWebConference.

Overall the research seems to advance the information discovery field on the web, and provides good contribution to the field of document labeling and classification.

Minor comments:
- Page 2, Line 122: "label" --> labels
- Page 4, Line 355: "dispenses" --> dispense?
- Page 8, Line 924: "outperforms" --> outperforming?

**Questions:**

- How can you make the paper more aligned with the conference track?

**Ethics Review Description:**

-

**Reviewer Confidence:**

3: The reviewer is confident but not certain that the evaluation is correct

**Scope:**

3: The work is somewhat relevant to the Web and to the track, and is of narrow interest to a sub-community

---

### Official Review · Reviewer_Xw6V · 2023-11-30

**Novelty:** 5
**Technical Quality:** 5

**Review:**

The paper is the frame of multi-label document classification and specifically tackles the problem of rebel completion/refinement.
The work specifically tackles the setting where the labels belong to a taxonomy, therefore the completion/refinement for each input label can be defined as a path.

The solution employs a transformer on a seq2Seq task and directly consumes information from the target taxonomy.

The paper is well written and motivated, and tested against state-of-the-art methods on 3 large datasets, obtaining promising results.

**Questions:**

- when producing the results, do you output a single sequence/path per document or one path per each input initial label?
- can the method be applied in the absence of initial labels?

[questions anwered by the authors]

**Reviewer Confidence:**

3: The reviewer is confident but not certain that the evaluation is correct

**Scope:**

4: The work is relevant to the Web and to the track, and is of broad interest to the community

---

### Decision · Program_Chairs · 2024-01-22

**Decision:**

Accept (Oral)

**Comment:**

This paper presents an approach to complement label predictions by exploiting hierarchical information between terms.
 The work addresses an important issue, the proposal is innovative and it has been quite appropriately evaluated.
 There are some noticeable suggestions for improvement, though. For example, the authors should make it more explicit, why the paper is relevant for this conference. This is not a big challenge, as the topic is indeed relevant (but not phrased as clearly as it could) and the authors seem to have understood what to do here. The same applies to other comments and questions which authors did not seem to have much issue answering.